# Enhanced Performance of a Visible Light Detector Made with Quasi-Free-Standing Graphene on SiC

**DOI:** 10.3390/ma12193227

**Published:** 2019-10-02

**Authors:** Xiaomeng Li, Xiufang Chen, Xiangang Xu, Xiaobo Hu, Zhiyuan Zuo

**Affiliations:** 1State Key Laboratory of Crystal Materials, Shandong University, Jinan 250100, China; lixiaomeng@mail.sdu.edu.cn (X.L.); xbhu@sdu.edu.cn (X.H.); zuozhiyuan@sdu.edu.cn (Z.Z.); 2Collaborative Innovation Center for Global Energy Interconnection (Shandong), Jinan 250061, China; 3Advanced Research Center for Optics, Shandong University, Jinan 250100, China

**Keywords:** graphene, SiC, optical detection

## Abstract

The excellent optoelectronic properties of graphene give it great potential for applications in optical detection. Among the graphenes obtained through many synthetic methods, epitaxial graphene obtained by thermal decomposition on silicon carbide has remarkable advantages for preparing photodetectors. In this research, epitaxial graphene has been successfully prepared on a silicon surface (0001) of semi-insulating 4H-SiC substrate with a size of 10 mm × 10 mm and epitaxial graphene has been converted to quasi-free-standing graphene by hydrogen passivation. Two metal-graphene-metal photodetectors were fabricated using the two types of graphenes above and the photo-absorption properties of detectors have been investigated under 650-nm laser illumination with different illumination powers. From a comparison of the performances between the two detectors, it was found that a photodetector fabricated with quasi-free-standing graphene shows enhanced performance under a light power of 0.018 mW. Responsivity and external quantum efficiency reach maxima of 5.11 A/W and 9.74%, respectively. This dramatic improvement is mainly due to the disappearance of the buffer layer in epitaxial graphene, providing a new method to achieve optimization of graphene-based opto-electrical devices.

## 1. Introduction

Photodetectors, part of the technical foundation of our information society, are widely used in communications, imaging, sensing and other applications [1]. Many different photodetectors, with different materials and structures, have been reported in recent years [2,3,4,5]. For these different photodetectors, the range of light detection depends on the light-absorbing materials, whose ability to absorb light is determined by their band structures. When the photon energy of the incident light is greater than the band gap of the photoactive material, large quantities of photo-induced carriers are generated and an electric signal is detected under a potential difference in an external circuit. Due to the increasing demand for high-performance miniature photodetectors in recent years, low-dimensional materials, because of their excellent performance, have received extensive attention and photodetectors with thin-film heterojunction structures have been studied in depth [6,7,8,9,10]. In these photodetectors, the built-in field at the interface of the heterojunction structures plays a very important role. The photo-induced carriers generated in the photoactive material can be rapidly separated by the built-in electric field so that a fast and strong photo response can be achieved. Therefore, heterojunction structures are widely used in various types of photovoltaic devices, such as solar cells and photodetectors, as a way to enhance photoelectric conversion efficiency [11,12]. As an exemplar of two-dimensional materials, graphene is widely used in photodetectors. But there is little research on photodetectors in which graphene itself is used as the light-absorbing layer. Because of the special band structure of graphene, it is more often used as a conductive layer in the fabrication of photodetectors, so that its optical absorption properties for visible light are not used. Therefore, using graphene as an absorbing layer in the detector deserves attention.

In the idealistic case, graphene is a two-dimensional material formed by SP^2^-hybridized carbon atoms arranged in a hexagonal honeycomb lattice. Thanks to its excellent optical and electrical properties, including wavelength-independent absorption [13], high thermal conductivity [14], low dissipation rate, extremely high carrier mobility [14] and the possibility to easily tune the carrier density [15,16], there have been many studies of photodetectors made with graphene [17,18,19]. However, since the band gap of intrinsic graphene is zero, a semiconductor substrate always acts as a light-absorption layer in conventional graphene/semiconductor photodetectors [20]. However, based on studies using band-gap manipulation of graphene, graphene itself can be used as a light-absorbing layer for visible to infrared bands. It has been reported recently that a few layers of graphene, with an appropriate driving voltage, can be highly responsive to infrared light [19]. This shows that band-gap manipulation of graphene is of great significance for its light-sensing applications.

There are many ways to manipulate the band gap of graphene. In particular, three mechanisms have been proposed—(1) Quantum confinement. Graphene is prepared as a graphene nano-strip; then quantum effects and edge effects cause the band gap of the graphene to be opened [21]; (2) Doping method. Atomic doping and adsorption cause the Fermi level of graphene to move up and down at the Dirac point, which contributes to the opening of the band gap [22]; and (3) Symmetry breaking. The symmetry of AB-type stacked graphene is much less than that of AA type stacked graphene. When different electric fields are applied to the bottom of its conduction band and the top of its valence band, the energy levels can be split and the energy band can be opened [23]. These methods make it possible to use graphene as an absorbing layer in a photodetector. Based on the above method (3), Chen et al. reported that the energy band of epitaxial graphene on SiC is not zero but is about 200 meV [24]. This is derived from the asymmetry of the sub-lattice in epitaxial graphene made by thermal decomposition method. Compared with other methods for manipulating the band gap of graphene, the thermal decomposition method on silicon carbide can avoid mechanical damage and chemical pollution in graphene and ensure its stability. We conclude that epitaxial graphene on SiC is very suitable as a light absorbing layer to be used in photodetectors.

In this research, metal-graphene-metal (MGM) photodetectors have been fabricated in which the two gold electrodes are the source and the drain, respectively, on the same side of the epitaxial graphene. Many detectors with various architectures have been reported, among which the MGM photodetector is the simplest. At the interface between graphene and metal, an electric field is generated due to carrier migration [25], so that the working area of the photodetector is limited to the contact interfacial regions.

Two kinds of epitaxial graphene for photodetector fabrication had been prepared. We named one of the graphenes, the one subjected to hydrogen passivation, quasi-free-standing graphene (QFSG). The other graphene, which was not subjected to hydrogen passivation, epitaxial graphene (EG). The main difference between these two graphenes is that there is a buffer layer in the EG and no buffer layer in the QFSG. Our experiments show that quasi-free state graphene on silicon carbide has better photoelectric conversion properties than epitaxial graphene. For photodetector fabricated with QFSG, a responsivity of 5.11 A/W and an external quantum efficiency of 9.74% at a wavelength of 650 nm are realized. These parameters are well-suited for graphene photodetectors.

## 2. Materials and Methods

### 2.1. Preparation of EG and QFSG

The epitaxial graphene (EG) for photodetector fabrication was obtained by thermal decomposition on the Si-terminated surface (0001) of semi-insulating 4H-SiC substrate with a size of 10 mm × 10 mm. The 4H-SiC substrate was single crystal and was supplied by State Key Laboratory of Crystal Materials, Shandong University. The substrate was processed using a chemical-mechanical polishing (CMP) technique in advance. Then a standard RCA cleaning method was performed. And RCA was a typical wet chemical cleaning method that was the most commonly used. The growth process of EG includes three principal steps—(1) The substrate is etched in order to obtain parallel and regular wide steps on the surface under a hydrogen atmosphere at a temperature of about 1600–1700 °C. (2) The graphene growth process starts when the temperature and chamber pressure reach about 1750 °C and 800 mbar, respectively. (3) The epitaxial graphene is cooled down to room temperature.

QFSG is obtained by extra hydrogen passivation treatment of EG. The silicon-carbon covalent bond between the buffer layer and the SiC substrate is broken and the silicon dangling bonds are saturated when the hydrogen passivation process is carried out [26]. This process was carried out by annealing in a hydrogen atmosphere at about 1000 °C. In this study, these two different types of epitaxial graphene were identified by atomic force microscopy (Dimension Icon; Vecco Dimension Icon, New York City, North America), scanning electron microscope (SEM) and X-ray photoelectron spectroscopy (XPS). In addition, Raman spectroscopy measurements were employed to determine the thickness of the graphene.

### 2.2. Fabrication of the MGM Photodetector

The application of traditional lithography technology could significantly damage the photoelectric properties of graphene. Therefore, in this study, electrodes were made by a sputtering process on two different epitaxial graphenes with a hollow aluminum contact pattern mask. Before the fabrication of the photodetector, these two epitaxial graphenes were cleaned using acetone and ethanol solutions. Then the surfaces of EG and QFSG were covered by aluminum masks, following which the sputtering processes were carried out. At the same time, gold clusters were regularly deposited on the surface of the epitaxial graphene to form gold films as dictated by the shapes of the aluminum masks. A structural diagram of the device sample is shown in Figure 1. And the digital microscopic image of actual device can be got from Appendix A. It can be seen that the gap between the electrodes has a light absorption area of about 1.3 mm^2^, while the total area of the device sample is about 4 mm^2^. A semiconductor test system (Keithley 4200A, Tektronix, Shanghai, China) was used in the performance measurements of the photodetectors to investigate the photodetecting properties of the two samples. The excitation light source came from a semiconductor laser diode system with a peak wavelength of 650 nm and a laser spot diameter of about 2 mm. A bias increase from 0 to 10 V was applied linearly to both electrodes. A photo response could be detected when the laser illuminated the device.

## 3. Results and Discussion

In this research, high-quality and micropipe-free SiC wafers were used to process the EG and QFSG. After the hydrogen etching was completed, we got a formation of regular steps morphology with micrometer-sized terrace width and elimination of surface defects on the SiC substrate. This is positive for nucleation and growth of high-quality epitaxial graphene. Subsequently, the carbon atoms preferentially nucleated at the edge of the step and then extended to the stepped platform. Large flat terraces on the SiC substrate can be seen in Figure 2a. It is clear that there are no defects on the order of micrometers on the SiC substrate or on the step. The images Figure 2c,f show the work function of the two kinds of epitaxial graphene (EG and QFSG). These were measured by kelvin probe force microscope (KPFM) in the white rectangular area in Figure 2b,e that contains complete steps. The work function of QFSG is significantly larger than that of EG. It can be inferred that the buffer layer was removed by hydrogen passivation treatment, resulting in the disappearance of n-type doped graphene and a decrease of the work function [27,28]. This can also be confirmed by the SEM image (Appendix A) of the two samples that the signal of the QFSG sample (Appendix A) with larger work function in the SEM image is weaker than the EG sample (Appendix A). Besides, in order to further verify the effect of hydrogen passivation on the buffer layer, we measured the two epitaxial graphenes (EG and QFSG) with X-ray photoelectron spectroscopy (XPS). As Figure 3 shows, compared with the XPS results of the EG (Figure 3a), the S1 and S2 peaks induced by the interaction between the buffer layer and the substrate were absent from the XPS results of the QFSG (Figure 3b). It can be concluded that hydrogen atoms have been sufficiently inserted between the graphene and the substrate, so that the buffer layer is completely converted into a graphene layer at a suitable temperature [29].

Additionally, Raman measurements were performed to characterize the layer thickness and the quality of the EG and QFSG. It is well known that there are three characteristic peaks in graphene’s Raman spectra, which are commonly called the D, G and, 2D peaks. They appear, respectively, around 1350, 1583 and 2700 cm^−1^. The intensity of the D peak is related to the degree of defects in graphene, so the D peak is also called the defect peak. The G peak is induced by the stretching vibration of the SP^2^ carbon atoms, so the stress applied to the graphene can be judged by the shift of the G peak position. As to the 2D peak, its relative intensity and full width at half maximum (FWHM) are usually used to estimate the number of layers of stacked graphene [30]. To estimate the quality of our graphene sample, we selected nine random points on the sample for Raman testing and the distribution of the test points is shown in the Appendix A. The Raman spectroscopy test results of the EG and the QSEG samples are shown in Figure 4a,b, with the background SiC spectra subtracted out. The characteristic G and 2D peaks of the two epitaxial graphenes (EG and QFSG) can be clearly seen. The D peak is hardly visible, meaning that graphene with a low density of defects has been obtained. By fitting with a single Lorentizian symmetrical peak, we got full width at half maximum (FWHM) of the 2D peak. The fitting results show that the FWHMs of EG ranged from 40 to 65 cm^−1^ and the FWHMs of QFSG ranged from 30 to 40 cm^−1^. By substituting the FWHMs of the 2D peak into the empirical formula.
(1)$$FWHM\left(2D\right)=\left(-45\left(1/N\right)+88\right){\mathrm{cm}}^{-1}$$

We can further find that the average number of layers of the EG and the QFSG are 1.38 and 0.38, respectively. The Raman data for two epitaxial graphenes show characteristics of the graphene structure with a buffer layer and with a graphene monolayer.

Photocurrent is a key parameter to characterize the photo-detection ability of photodetectors made with the two epitaxial graphenes. The photocurrents of the two samples were obtained by 650-nm-wavelength laser illumination with different light powers, as shown in Figure 5a,b. Both of these curves of the two samples are plotted as averages of the data from three repeated tests (all repeated test data for each sample can be found in our Appendix A). It can be clearly seen that the photocurrent value of the QFSG photodetector illumination by the laser is higher than the current when there is no laser illumination. But the situation is different for the EG photodetector. The photocurrent of the EG photodetector excited by the laser is lower than the current when the photodetector is not illuminated. This can be attributed to the strong scattering introduced to the epilayer caused by the existence of buffer layer, which makes the mobility of the epilayer strongly temperature-dependent and dramatically limits its room-temperature mobility [24]. Net photocurrent is a more useful parameter for characterizing the performance of a photodetector. It can be expressed by:(2)$$I=I_{ph}-I_{dark}$$
where *I_ph_* is the photocurrent excited by the laser and *I_dark_* is the current observed without illumination. Here the net photocurrent curves of the QFSG sample are obtained by changing the bias voltage from 0 up to 10.0 V, as shown in Figure 5c. It is derived from the average of the data from three repeated tests on the same sample and it reveals that the net photocurrent of the device decreases with an increase of light power, which results from the thermal radiation and the stronger scattering between excited electrons when the light power increases [31]. If the device is to be applied in photodetection, this behavior needs to be adjusted by temperature control [32]. The net photocurrent reaches a maximum value of 2.38 × 10^−5^ A at a laser power of 0.018 mW and a bias voltage of 10 V. When the laser power exceeded 0.036 mW, there were some different results—the photocurrent obtained under the 0.051 mW laser irradiation was larger than that obtained under the 0.075 mW laser irradiation. That can be related to the heat capacity of graphene and Seebeck effect, which needs to be studied more deeply (the results of three repeated tests are shown separately in Appendix A). There are many deviations from smoothness in these curves, most likely originating from the changes in environmental conditions during testing. At the same time, it was found that the net photocurrent of the device decreased as the number of repeated tests increased. This shows that the durability of the device needs to be strengthened.

In contrast to the photocurrent test results of the QFSG (Figure 5a), *I_ph_* of the EG sample was smaller than *I_dark_* that was shown in Figure 5b. This means that no net photocurrent was generated when the laser illuminated the EG sample. However, the net photocurrent could be measured in the third repeated test Appendix A when the optical power was, successively, 0.018 mW, 0.036 mW and 0.051 mW. That can be summarized as originating from the instability of the test environment conditions. In summary, the existence of the buffer layer introduces strong scattering to the epilayer, which makes the mobility strongly temperature dependent and limits its room-temperature mobility. Therefore, under the same test conditions, the net photocurrent of the EG sample is lower than that of the QFSG sample [24].

As two important parameters for detector performance, the responsivity (R) and external quantum efficiency (EQE) of the QFSG sample in this experiment were also calculated. R can be obtained by substituting the photocurrent value into the following formula:(3)$$R=\frac{I}{P_{irra}\times S_{eff}}$$
where *P_irra_* is irradiation power and *S_eff_* is the effective area of the detector. Another important parameter used to evaluate detector sensitivity to photons can be expressed as
(4)$$\mathrm{EQE}=\frac{R\times hc}{e\lambda }$$
where *e* is the elementary charge, *λ* is the incident laser wavelength, *h* is Planck constant and *c* is the speed of light. *R* and *EQE* of the QFSG sample obtained with 650-nm laser irradiation are shown in Figure 5d, where EQE is proportional to responsiveness. Following trends similar to those of the net photocurrent, values of *R* and *EQE* decrease as the light power increases when the QFSG sample is illuminated by 650-nm laser light. From Figure 5d, we see that the maximum values of *R* and *EQE* at a light power of 0.018 mW are 5.11 A/W and 9.74%, respectively. Surprisingly, these results, obtained for the QFSG photodetector in this research, are significantly higher than those previously reported for graphene photodetectors [25,33].

## 4. Conclusions

It can be concluded from the above comparison that the net photocurrent of the QFSG sample is significantly higher than that of the EG sample. The Fermi level of graphene in the EG sample was increased by the underlying buffer layer, resulting in the conversion of graphene to n-doped graphene [28]. That caused more scattering to occur during the propagation of photo-generated carriers, hindering the photoresponse of the EG sample. In addition, as shown in Figure 2c,f, the work function of graphene in the EG sample is lower than that in the QFSG sample. These results in a higher barrier between the graphene and the Au electrode in the EG sample than in the QFSG sample, which also made the net photocurrent in the EG sample more difficult to detect.

In summary, the buffer layer in epitaxial graphene on SiC severely limits its photo-absorption property. Here we used the hydrogen passivation method to get QFSG from EG and we manufactured an MGM photodetector using QFSG. Photoresponse testing for 650-nm-wavelength lasers was implemented on the QFSG sample, giving 5.11 A/W and 9.74%, respectively, for the maximum values of R and EQE at a light power of 0.018 mW. These results show that QFSG has great potential as a visible-light photodetector and deserves further development.

## Figures and Tables

**Figure 1 materials-12-03227-f001:**
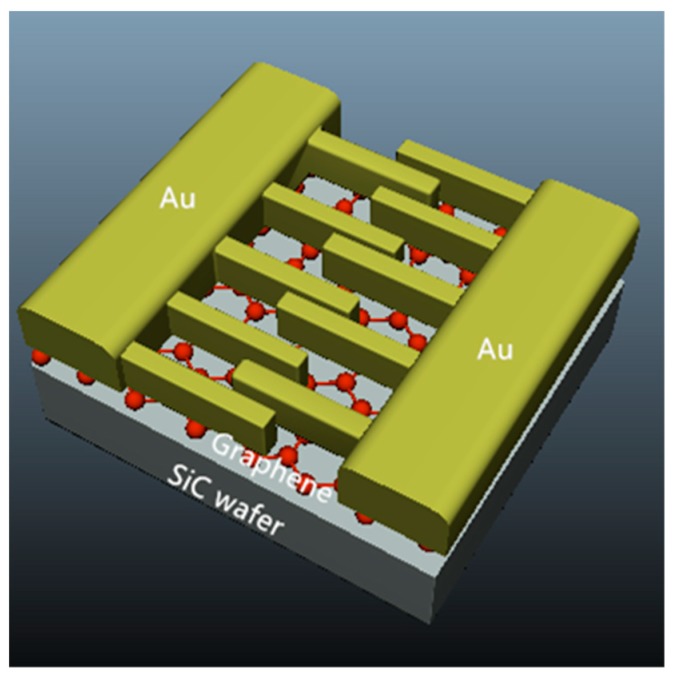
The 3D structure diagram of epitaxial graphene photodetector, the light absorption area is about 1.3 mm^2^ while the total size of the device is about 4 mm^2^.

**Figure 2 materials-12-03227-f002:**
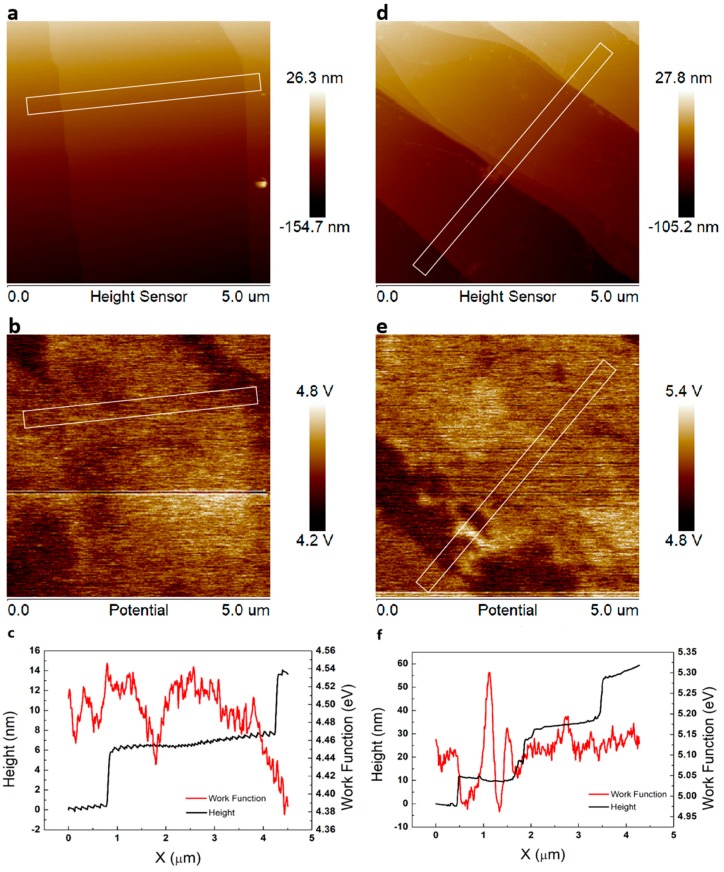
The atomic force microscopy (AFM) image of the epitaxial graphene (EG) sample (**a**) and quasi-free-standing graphene (QFSG) sample (**d**) within a 5 × 5 square micron square. The steps on the surface of both samples are on the order of micrometers with no obvious defects. (**b**,**e**) are respectively the Kelvin probe force microscope (KPFM) test results of the EG sample and the QFSG sample. And the height distribution and work function distribution of EG sample (**c**) and QFSG sample (**f**) were obtained from the white rectangular area in (**a**,**b**,**d**,**e**). The work function of the EG sample is about 4.50 eV. The work function of the QFSG sample is about 5.15 eV.

**Figure 3 materials-12-03227-f003:**
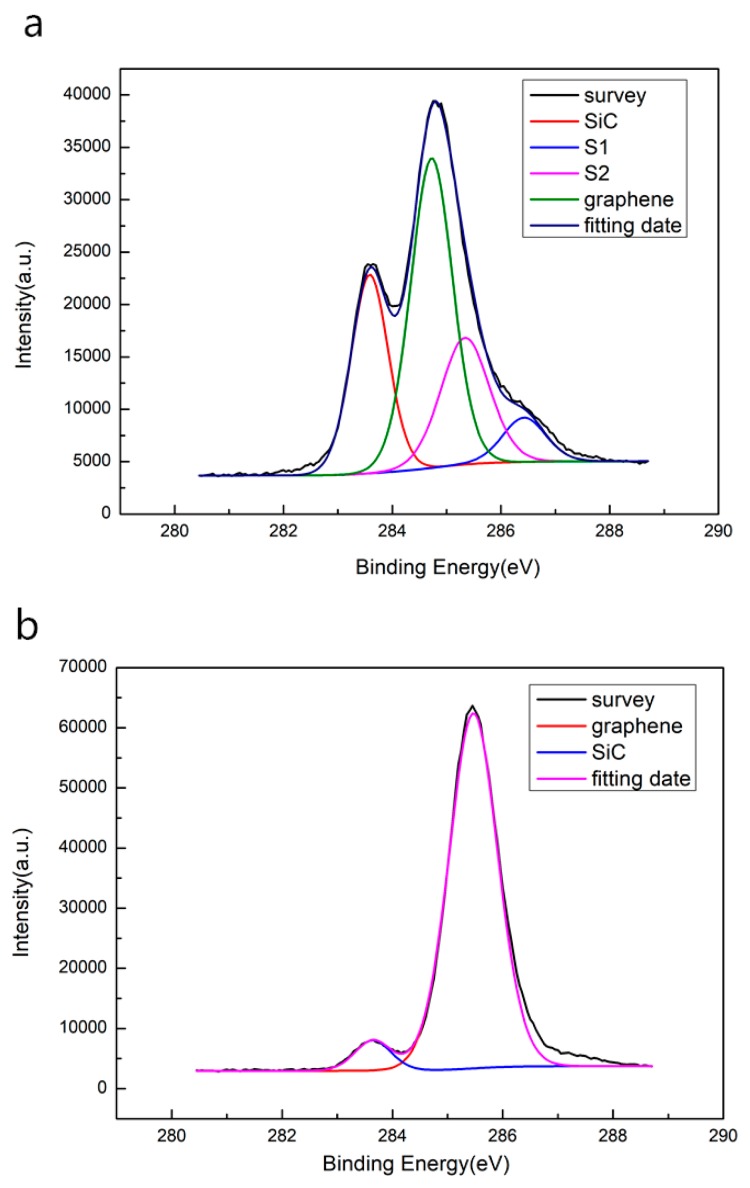
X-ray photoelectric spectroscopy of EG sample (**a**) and QFSG sample (**b**).

**Figure 4 materials-12-03227-f004:**
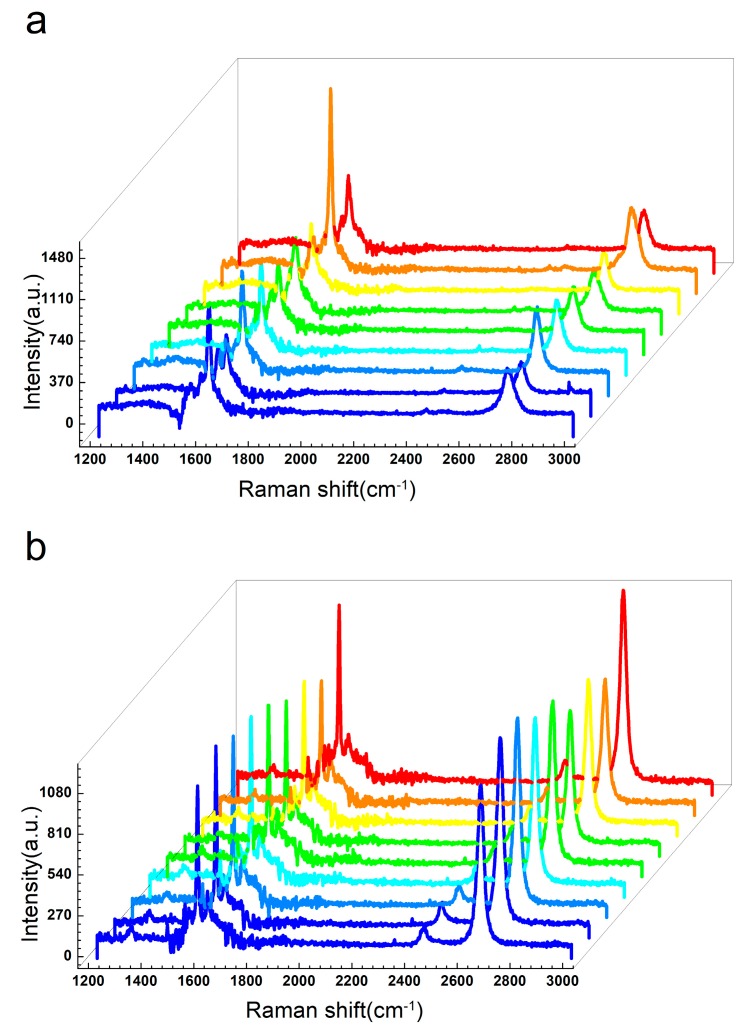
Raman spectra at nine random points on the surface of EG sample (**a**) and QFSG sample (**b**).

**Figure 5 materials-12-03227-f005:**
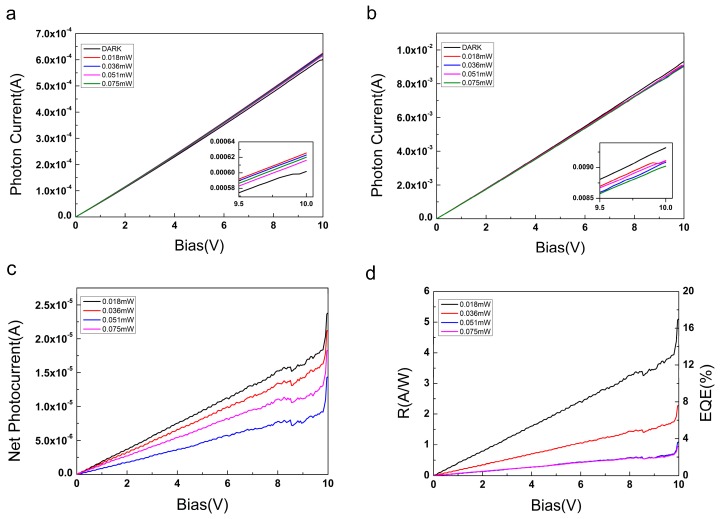
(**a**) Photon currents of QFSG samples, obtained with and without illumination, the excited wavelength is 650nm and the laser power is provided in the figures. The details of the data are magnified and displayed in the inset. (**b**) Photon currents of EG samples, obtained with and without illumination. The details of the data are magnified and displayed in the inset. (**c**) Net photocurrents of QFSG sample, reported as a function of the bias voltage, for different values of the light power. (**d**) Responsivity and external quantum efficiency of QFSG sample, reported for different values of light power.

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
