# Peer review of "Enhanced Performance of a Visible Light Detector Made with Quasi-Free-Standing Graphene on SiC"

_materials, 2019, doi:10.3390/ma12193227_

Round 1

Reviewer 1 Report

The Authors fabricate and experimentally analyze the performance of a visible light MGM photodetector based on a graphene layer, obtained by thermal decomposition on SiC. Comparing the results obtained for an epitaxial graphene and for a quasi-free-standing graphene, they otain no net photocucurrent for the first one, and a good performance for the second one, due to the lack of the buffer layer.

The paper is well-written and interesting, but some points should be clarified and improved.

In particular:
1) I suggest to increase the size of the figures, and in particular, of the labels of the axes, which at the moment are very difficult to read (moreover, at the moment the supplementary material is inaccessibile).
2) At page 6, the Authors report a strange behavior: increasing the light power the net photocurrent decreases. If I have understood well, the Authors justify this saying that increasing the light power, even though the number of generated electron-hole pairs increases, the net photocurrent decreases because of the increased transport scattering. In my opinion, this is an unwanted behavior and this should be explicited in the text.

Other minor points to improve are the following:
3) At page 2, among the advantages of graphene for the photodetector fabrication, it should be cited also the possibility to easily tune the carrier density in the graphene (I suggest to cite your reference [12], together with:

P. Marconcini, M. Macucci, "Approximate calculation of the potential profile in a graphene-based device," IET Circuits, Devices & Systems 9, 30 (2015), DOI: 10.1049/iet-cds.2014.0003).

Q. Cui, Y. Yang, J. Li, F. Teng, X. Wang, "Material and Device Architecture Engineering Toward High Performance Two-Dimensional (2D) Photodetectors," Crystals 7, 149 (2017), DOI: 10.3390/cryst7050149 

4) Page 2, line 82: I would move "we call" at the beginning of the sentence;
5) Page 2, lines 56 and 82: change "grapheme" to "graphene"
6) Page 3, lines 114 and 115 (but also page 5, lines 158, 168, 169): correct the position and size of the exponent;
7) Page 3, line 122: change "photodetector with" to "photodetector;"
8) Page 4, lines 135, 136, 148, 149: change "um" to "micron"
9) Page 4, line 146: substitute the comma with a dot
10) Page 5, line 154: write better the sentence: for example substitute "calculated" with "characterized", complete the sentence for the (b) panel
11) Page 5, line 171: add a bracket
12) Page 5, line 172: does "0.38" mean that most of the times in the selected points there is no layer?
13) Page 5, lines 180-181 (but also page 5, lines 201-202): remove the semicolon (and complete the bracket)
14) Page 6, line 194: change "It" to "it"
15) Page 6, line 199: change "larger" to "smaller" (is it right?)
16) Page 6, line 206: change "5(b)" to "5(c)"
17) Page 7, lines 231-234: improve the sentence (for example: "a) Photon currents of QSFG samples, obtained with and without illumination, the excited wavelength is 650nm. b) Photon currents of EG samples, obtained with and without illumination. c) Net photocurrents of QSFG sample, reported as a function of the bias voltage, for different values of the light power. d) Responsivity and external quantum efficiency of QSFG sample, reported for different values of light power.)
18) Page 9, line 312: write better "SiO2" in the title of the reference

Author Response

Dear Reviewer,

We would like to thank Materials for giving us the opportunity to revise pur manuscript. And thank you for your careful read and thoughtful comments on previous draft. We have carefully taken your comments into consideration in preparing our revision, which has resulted in a paper that is clear, more compelling and broader. The Submitted document contains our response.

Thanks for all the help!

Best wishes,

Xiaomeng Li

Author Response

Dear Reviewer,

We would like to thank Materials for giving us the opportunity to revise our manuscript. We thank you for your careful read and thoughtful comments on previous draft. We have carefully taken your comments into consideration in preparing our revision, which has resulted in a paper that is clearer, more compelling and broader. The attachment contains our response.Please see the attachment.

Thanks for all the help.

Best wishes,

Xiaomeng Li

Reviewer 3 Report

The authors present an interesting approach of using graphene deposited on 4H SiC as possible photodetector. The study's principle is well-designed and may be of potential interest to the audience of Materials, but there are nevertheless serious flaws throughout whole manuscript. Quality of data presentation is also very low and in order to consider this manuscript for publication. In many cases the assumptions are not backed by the literature, which seriously impairs the overall scientific soundness. There are also several scientific mistakes that have to be addressed. The summary of comments indicating the major inconsistencies is listed below and the authors are encouraged to address these issues.

Line 1: Please specify the type of journal.

Line 48, sp2 should go to the superscript. Additionally, sp2 is an idealized case, while the surface of graphene is rich in oxidized species. Pleasse specify that you refer to an idealistic case.

Line 82- typo "grapheme"

Line 84- what is meant by buffer layer here? The carbon layer between the graphene monolayer and SiC? Please specify.

Line 87- "these parameters are well-suited for graphene photodetectors"- based on what assumptions.

Line 99-100- the sentence about the dangling bonds of Silicon being saturated- please provide a reference.

Line 114 and 115- please put the quadratic term into the superscript

Line 113: why is the digital microscopic image actual device not shown?

Line 127: Was the 4H SiC single crystal? Supplier?

Line 128:

Figure 2: The quality of the figure is not satisfactory. It lacks the structure (a,b,c are of different sizes and look like snapshots from the measurements). Additionally the height sensor on the AFM a) is not scaled! The authors also do not indicate why the deflection signal is displayed by the height signal.

Fig 3: EG sample is calculated by XPS. Do the authors mean "from" the XPS? Please provide details on peak deconvolution and baseline generation. Please rewrite the descrition of the figure.

Line 159: This is not true, D stands for diamond-like carbon peak, whereby the deffects can be read out from the distribution of the 2D peak.

Fig 4: Please restructure the data. Plots hardly visible. Export the plots in at least 300 dpi and please pay attention to the unified format.

Line 178: no laser power provided.

Line 200: This effect can be related to the heat capacity of graphene and Seebeck effect.

Line 238-239: "resulting in the conversion of graphene to N-doped graphene"- authors probably mean n-doped graphene? Otherwise N (capital letter) reads as nitrogen here. Please provide explanation why the n-doping is assumed.

Fig 5: Hardly visible graphs. It is almost impossible to distinguish between the curves.

Author Response

Dear Editor,

We would like to thank Materials for giving us the opportunity to revise our manuscript. We thank you for your careful read and thoughtful comments on previous draft. We have carefully taken your comments into consideration in preparing our revision, which has resulted in a paper that is clearer, more compelling and broader. The attachment contains our response. Please see the attachment.

Thanks for all the help.

Best wishes,

Xiaomeng Li

Round 2

Reviewer 2 Report

n/a

Author Response

Dear Reviewer,

We thank you for your careful read during the second review. No intention to offend, but your comment is invisible

Thanks for all the help.

Best wishes,

Xiaomeng Li

Reviewer 3 Report

I would like to thank the authors for adapting to the suggested changes and their revision of the manuscript.

Unfortunately I still see serious flaws from the scientific point of view:

Fig 2 b) and c). AFM (I assume tapping mode) results are not compensated, nor normalized to the 0 nm value. Also the scans look like during the scan the integral/ proportional gain were not adjusted. It has resulted in apparently well resolved image, which nevertheless does not take any features into consideration. And sorry, but 2 d) looks like a systematic drift.

Fig 4) a) and b). Why are the Raman spectra shown as waterfall? The reason of using random points on the structure (of which SEM is still not provided- even if the sample is not conductive- that should be possible at low acceleration voltages) is not of highest scientific soundness, unfortunately.

The results were on the other hand updated and well described. I suggest the following:

Please combine Fig. 5 into a), b) c) d) insets in 4 quadrants as one image.

Same for the Fig 2 and 3 (make sure that all the scans are of same area please, meaning e.g. 5 x 5 um). Inset d may be even omitted.

Please provide SEM image of the device

Please describe all the axes on the Fig. 4. 

The results might be interesting to the audience, but without serious consideration of the comments and indications above the best I can do is to recommend the article for major revision. 

Author Response

Dear Reviewer,

We thank you for your careful read and thoughtful comments during the second review. We have carefully considered your comments and modified my manuscript based on them, which makes my manuscript more rigorous. And the attachment contains our response. Please see the attachment.

Thanks for all the help.

Best wishes,

Xiaomeng Li
